# Nitrogen Fertilization and Cultivar Interactions Determine Maize Yield and Grain Mineral Composition in Calcareous Soil under Semiarid Conditions

**DOI:** 10.3390/plants13060844

**Published:** 2024-03-14

**Authors:** Ivica Djalovic, P. V. Vara Prasad, Kashif Akhtar, Aleksandar Paunović, Muhammad Riaz, Marijana Dugalic, Snežana Katanski, Sajjad Zaheer

**Affiliations:** 1Institute of Field and Vegetable Crops, National Institute of the Republic of Serbia, 11060 Novi Sad, Serbia; snezana.katanski@ifvcns.ns.ac.rs; 2Department of Agronomy, Kansas State University, Manhattan, KS 66506, USA; 3State Key Laboratory for Conservation and Utilization of Subtropical Agro-Bio-Resources, College of Life Science and Technology, Guangxi University, Nanning 530004, China; kashif@zju.edu.cn; 4Faculty of Agronomy, University of Kragujevac, 32000 Cacak, Serbia; aleksandar.s.paunovic@gmail.com; 5Department of Environmental Sciences, Government College University Faisalabad, Allama Iqbal Road, Faisalabad 38000, Pakistan; mr548@ymail.com; 6Faculty of Agriculture, University of Nis, Kosančićeva 4, 37000 Kruševac, Serbia; marijanadugalic80@gmail.com; 7Department of Agronomy, Faculty of Crop Production, The University of Agriculture, Peshawar 25130, Pakistan; sajjadzaheer15@gmail.com

**Keywords:** maize, fertilization, mineral composition, yield, South Pannonian Basin

## Abstract

Identifying the contributions of climate factors and fertilization to maize yield is significant for the assessment of climate change impacts on maize production under semiarid conditions. This experiment was conducted with an overall objective to find how N fertilization and cultivar interactions along with climatic conditions determine the mineral composition and maize yield responses of four divergent maize cultivars grown under eight different fertilization levels. The results showed that element contents were significantly affected by year (Y), cultivar (C), N fertilization, and N × C interaction. The element contents of grains were mainly influenced by N rate or N × C interactions. The results showed that maize yield was significantly affected by year (Y), genotype (G), N fertilization (N), and Y × G × N interaction. These results implied that the maize yield was significantly affected by changes in genotypes and environments. Overall, our findings are a result of the interactions of genetic, environmental, and agronomic management factors. Future studies could evaluate more extreme plant densities, N fertilizer levels, and environments to further enhance our understanding of management effects on the mineral composition and maize yield in calcareous soil.

## 1. Introduction

Maize (*Zea mays* L.) is one of the main crops in the world and grown in more than 130 countries. Maize is one of the main crops grown for human food and livestock feed, using more than 8.5 million ha of cropland annually in Europe [1]. Maize production is severely affected by abiotic stresses like low and high temperature, flooding, drought, low soil fertility (salinity, acidity), imbalanced nutrition, and micronutrient deficiency that generally limit its yield worldwide [2,3].

The exploitation of genetic potential is possible with the cultivation of genotypes suitable to different climatic conditions and the application and adoption of modern agro-technology in maize production. Among the new technologies of maize production enhancement, nutrient management is a key component. Efficient nitrogen (N) fertilizer management is essential for achieving economic yields and for enhancing N use efficiency [4]. Effective N management can improve N use efficiency (NUE) and depends largely on the combination of appropriate sources, application rates, timing, and the placement of N fertilizer application [5]. Many researchers have reported that under the same N application rate, splitting the number of N applications according to the plant’s N needs can reduce N losses and increase yield and NUE. Various nitrogen management practices, including split applications during planting and vegetative stages, have been found effective in improving maize yields and nitrogen use efficiency. Wang et al. [6] reported that the N recovery efficiency for the two-split and three-split application of urea was significantly higher than that for the one-time mixed basal dressing. Nitrogen fertilization (40% at the same time as sowing and 60% in the spring part of the vegetation in phase 8 leaves) achieved higher corn grain yield compared to the total amount of nitrogen applied in the spring and in fertilization [3]. These authors conclude that multiple applications of nitrogen are more desirable on production surfaces that contain less accessible nitrogen. Vetsch and Randall [7] stated that in the first year of the study, in which April and May were dry and warm with a mean daily temperature significantly higher than the perennial average, maize grain yield was 20% lower on nitrogen-treated treatments in the spring compared to the yield obtained on the variant where nitrogen was applied in the autumn, while in the other two years of the test, there were no significant differences in the yield obtained by applying nitrogen at different times. Adu et al. [8] showed that increasing nitrogen fertilizer causes an increased nitrogen concentration in some maize genotypes. Increasing the amount of nitrogen affects the absorption of other elements such as potassium, magnesium, calcium, and phosphorus and, in some cases, intensifies the absorption of some elements [9]. Applying more nitrogen to maize resulted in maximum emergence as well as improved plant elongation and yield [10]. Nitrogen fertilizer also increased maize grain production (43–68%) and biomass (25–42%) [11]. Responses in grain yield and fertilizer recovery to late N application are observed because modern high-yielding hybrids take up a considerable amount of N after flowering [12]. Therefore, as the N supply to grains via remobilization from leaves and the stem is limited, the uptake of high amounts of N is required during the reproductive stages [13,14]. In semiarid climates, growing maize hybrids is more efficient than local cultivars in terms of the higher NUE and grower’s income. However, no previous research was reported to investigate the response of different maize genotypes to different N sources × N levels interactions. 

This experiment was conducted with an overall objective to find how N fertilization and cultivar interactions along with climatic conditions determine the mineral composition and maize yield in calcareous soil under semiarid conditions.

## 2. Results

### 2.1. Variations in the Plant Nutrients (N, P, and K) of Different Maize Genotypes

Over two years, significant changes were observed in leaf and grain nutrient contents under various N fertilization (N) in different cultivars (C), while the interactive response of N × C was not significant for nutrients except leaf K during 2012 and was significant for grain N content in 2012 (Figure 1 and Figure 2; Table 1). 

Further, over two years, the responses of the treatments were significant for N and P content in leaves but not K content, and the effect of different cultivars showed significant performance in leaf N content (2011) and P content (2011 and 2012) but not K content. Moreover, the N content of grain was significantly affected during 2011 and 2012, while P content was significantly affected in 2011 under different treatments. The performance of different cultivars revealed that grain N, P, and K were significantly affected during 2012.

### 2.2. Effects on Leaf and Grain Micronutrients of Different Maize Cultivars

Over two years, the responses of different N fertilization (N) and cultivars (C) on leaf and grain micronutrients of maize are presented in Table 2 and Table 3.

The results showed that leaf Na was significantly (*p* < 0.05) affected by cultivars and was not significantly affected by N fertilization or the interaction of N × C. Further, grain Na was also found not significant for treatments, cultivars, or interaction. In addition, the leaf Na content was significantly 274 mg kg^−1^, increased by the NS-6030 maize cultivar over 2012. The leaf Ca, Mg, and Mn content was significantly (*p* < 0.05) affected by treatments and cultivars but did not have a significant response to the interaction of N × C. Meanwhile, the Ca content of the leaf NS-640 cultivar was also found not significant for treatments, cultivars, or N × C. The leaf Ca content significantly increased by 1.01 and 0.93 mg kg^−1^ in the N6-treated plot over two years. Meanwhile, the Ca content in the leaf of NS-4023 significantly (*p* < 0.05) improved by 0.97 mg kg^−1^ in the N5 plot and 0.84 in the N7-treated plot over two years. Similarly, significant Ca contents of 1.0 mg kg^−1^ by N8 and 0.91 mg kg^−1^ by N5 and N6 in the leaf of the NS-6010 cultivar were noted over two years. Further, the NS-6030 cultivar of maize showed a 0.91 mg kg^−1^ increase in the N6-treated plot during 2011, and a 0.90 and 0.91 mg kg^−1^ increase in the N5 and N6 plots over 2012. Maize cultivars NS-640, NS-4023, NS-6010, and NS-6030 showed improvements of 0.33 and 0.32 mg kg^−1^, 0.28 and 0.30 mg kg^−1^, 0.32 and 0.40 mg kg^−1^, and 0.26 and 0.33 mg kg^−1^ in the leaf by N6 treatment over two years. The statistical analysis showed that Mg content in the leaf was found significant for the treatment and cultivars, while not significant for the interaction of N × C. Zn content was found not significant for the treatment and interaction of N × C in 2011, but it was significant for the rest of the treatments, cultivars, and interactions. The Zn content in the leaf of NS-640 was 65.1 and 72.0 mg kg^−1^ and NS-4023 was 64.8, and 60.2 mg kg^−1^ was significantly improved by N7 treatment in 2011 and 2012. Further, N1 improved 61.9 and 48.5 mg kg^−1^ of the leaf Zn content of NS-6010 and NS-6030 maize cultivars in 2011, while the N7 treatment reported a significant increase of 59.6 and 46.8 mg kg^−1^ in the leaf Zn content of NS-6010 and NS-6030 maize cultivars over 2012. The Mn content in the leaf of NS-640 showed a significant increase of 90.9, 89.5 mg kg^−1^ by N7, and 90.5 in N6 over two years. Meanwhile, an 89.7 and 66.5 mg kg^−1^ increase in leaf Mn content was noted in the N5- and N7-treated plot of NS-4023 in 2011 and 2012. Further, NS-6010 and NS-6030 maize cultivars increased leaf Mn content by 88.9 and 87.9 mg kg^−1^ in 2011 and 75.5 and 93.5 mg kg^−1^ in 2012 in the N3-treated plots. Similarly, NS-640 and NS-4023 cultivars showed an increase in leaf Cu content by 19.7 mg kg^−1^ in N6 and 14.0 mg kg^−1^ in N5 treatment and 24.1 mg kg^−1^ in N7 and 13.9 mg kg^−1^ in N6 in the leaf of NS-4023 over two years. Further, 19.9 and 13.1 mg kg^−1^ and 18.4 and 13.9 mg kg^−1^ of leaf Zn content was significantly increased by N6-treated plots in 2011 and 2012. Moreover, the significant performance of the cultivar revealed that the higher amount of 274.5 mg kg^−1^ of Na was significantly increased by NS-6030 in 2012, while no significant relation was noted in 2011. Further, NS-640 reported that the Ca, Mg, and Zn content was significantly 0.95 and 0.88 mg kg^−1^, 0.30 and 0.30 mg kg^−1^, 49.3 and 44.9 mg kg^−1^, and 82.2 mg kg^−1^, and NS-6030 showed an 84.2 mg kg^−1^ increase in 2011 and 2012. Meanwhile, cultivar NS-6010 increased 17.3 and 12.8 mg kg^−1^ of the Cu content over two years. Micronutrient (Na, Ca, Mg, Zn, and Cu) contents in the grain of NS-640, NS-4023, NS-6010, and NS-6030 maize cultivars were not significantly affected by N fertilization, cultivars, or the interaction of N × C. Meanwhile, the Mn content in grain was significantly affected by cultivars over two years. The highest content of 0.95 and 0.88 mg kg^−1^ of Mn in the grain of NS-640 was noted over 2011 and 2012. Meanwhile, the highest Mn content (7.99 mg kg^−1^) in the grain of NS-6030 was reported during 2012 compared with the rest cultivars.

Further, Figure 3 shows that the Fe content in the grain of NS-640, NS-4023, and NS-6010 was significantly affected by N fertilization and cultivars and was not significantly affected by the interaction of N × C, and NS-6030 performance was found to be not significantly affected by treatments, cultivars, or the interaction of N × C. Moreover, the Fe content in the grain of NS-640 by N8 and NS-4023 and NS-6010 by N6 was significant over the two years.

### 2.3. Relationship of N, P, and K with Micro-Elements of Maize Genotypes over Two Years

Pearson’s correlation analysis showed that leaf Na, Ca, Mg, Zn, Mn, and Cu were significantly positively correlated with leaf N, P, and K and grain N content (Table 4), and a non-significant relationship was noted for leaf Mg and Cu with leaf K contents in 2011. Meanwhile, leaf Na, Ca, Mg, Zn, Mn, and Cu showed non-significant correlations with grain P and K contents during 2011. Negative correlations were noted between leaf Zn with leaf K and leaf Mn with grain K. Meanwhile, for 2012, Pearson’s correlation showed that leaf Na, Ca, Mg, Zn, Mn, and Cu had significant positive correlations with leaf N and P, except leaf Na with leaf N. While grain N content showed a significant relation with Ca, Mg, Zn, Mn, and Cu, a significant negative relationship between leaf Mn and grain K was noted in 2012 (Table 4). Further, Figure 2 shows a significant positive correlation of leaf N with grain N. 

### 2.4. Variations in Yield 

During the two-year period (2011 and 2012), the yield performance of four different maize cultivars was significantly affected by different treatments presented in Table 5.

The higher significant 12,934 kg ha^−1^ and 12,310 kg ha^−1^ yield of NS-640 maize cultivar was reported by N7 and N6 during 2011, while the yield of the same cultivar in the 2012 experimental year was 8357 kg ha^−1^ and 7991 kg ha^−1^ attained by N3 and N4 treatments. Further, the NS-4023 maize cultivar produced significant 11,231 kg ha^−1^ and 10,993 kg ha^−1^ yields in 2011 by N6 and N3 plots, while there was a 7620 kg ha^−1^ yield in 2012 from the N3-treated plot. In addition, the significant 12,502 kg ha^−1^ and 12,044 kg ha^−1^ yields of NS-6030 from N6 and N7 treatments and the 12,738 kg ha^−1^ and 12,163 kg ha^−1^ yield of NS-6030 were achieved from N7 and N6 during 2011, while N3 treatment reported an 8028 and 8317 kg ha^−1^ increase in the yields of NS-6010 and NS-6030 in 2012. However, Table 5 shows that N6 performed well for all maize cultivars except NS-640 in 2011, while the performance of the N3 treatment was approached for all maize cultivars in 2012; the variation in yield with respect to treatments in the second year of the experiment might be due to the change in climatic conditions. Further, a 10,621 and 7561 yield kg ha^−1^ were attained from the NS-6030 maize cultivar over two years. These results implied that the maize yield was significantly affected by changes in genotypes and environments.

## 3. Discussion

Maize (*Zea mays* L.) is one of the main crops worldwide and a significant source of various nutrients. Genetic, environmental, and agronomic management factors are widely used to increase crop growth and yield and to promote the sustainable production of the maize crop [3]. Optimizing N management can maximize N utilization by improving fertilization modes or modifying fertilizer types [15]. Modern maize genotypes reach high yields yet are susceptible to abiotic stress. Improving maize stress response is a great challenge due to the complexity of genotype (G) × environment (E) interactions, in particular during climate change [16]. Among abiotic factors, rate fertilizers, temperature, and precipitation are very important factors which dominantly affect the maize yield [2].

Maize belongs to the group of cultivated plants with the highest production of organic matter per unit area, so it is necessary to provide an adequate amount of nutrients to form high yields [17]. The maximum yield of cultivated plants can only be achieved by harmonious mineral nutrition, which is why it is recommended to always study the contents of as many elements as possible, sometimes even those that are not necessary, since the adequate and/or optimal security of certain elements is based on the absence of another element, or vice versa; the low content of a particular element may not always indicate its lack [18]. Different maize cultivars are able to exhibit different growth dynamics and organic matter production, as well as to accumulate different contents of mineral elements in specific environmental conditions, so that selection can create cultivars that are more efficient at absorbing individual nutrients [19]. The accumulation of some elements in a plant depends on the plant species and cultivar, physicochemical properties, pH value, organic matter content, cation exchange capacity, soil chloride content, N and P fertilizers, crop rotation, and previous crop [20,21,22,23]. Applying appropriate doses and NPK nutrient ratios, the adverse effect of drought can be mitigated, since it has been observed that a harmonious mineral diet reduces the transpiration coefficient of plants, that is, reduced water consumption for the synthesis of a dry matter unit [24,25].

Genotypic differences in the uptake and accumulation of individual elements between different genotypes are also reflected in their different reactions to fertilization, especially nitrogen [26,27,28,29,30]. The results showed that element contents were significantly affected by year (Y), cultivar (C), N rates (N), and N × C interaction. The element contents of grains were mainly influenced by N rate or N × C interactions. Environmental factors such as temperature and moisture also affect the availability of macro- and micronutrients [31]. Studies conducted in the corn belt of the USA [13,32], tropical regions [33,34], and Europe [27,35,36,37] show that genotypes can differ significantly in the level of N utilization. Some authors believe that in the period before flowering, there is competition between the reproductive and vegetative organs in assimilative uptake and that their accumulation in the grain depends on the amount of accumulated assimilative [38]. The accumulation of N in the reproductive period is often limited by unfavorable humid conditions, so the activity of the assimilation organs in such conditions is focused on reusing the amounts of N accumulated until then [39]. Late-maturing maize hybrids react more strongly to intensive N nutrition, because they have a longer vegetation period, i.e., a longer period of N absorption from the soil. Glass et al. [40] stated that the uptake of nitrate or ammonium ions depends on their concentration in the nutrient solution and that their concentration in plants increases with the increase in applied N doses [41]. Bundy [42] concluded that fall N application is an acceptable option on medium to fine-textured soils where winter temperatures retard nitrification. However, under these conditions, fall-applied N is usually 10 to 15% less effective than spring-applied N. The relative efficiency is primarily determined by the physicochemical properties of the soil and the agroecological conditions of a specific area, as well as the location and year [43]. Some authors point out that phosphorus fertilization is most effective in dry years, because in years with a precipitation deficit, the utilization of P by plants is reduced to a greater extent than N, which causes a higher proportion of soluble fractions of N in the plant [44]. The lack of P in dry conditions affects the reduction in root growth; the aerial part is more difficult to supply water, chlorosis appears, as well as the extinction of secondary shoots and the thinning of crops, which affects the yield’s reduction. The lack of micro-elements is becoming a growing problem of global proportions, since the intensive cultivation of high-yielding hybrids, with the application of higher doses of nitrogen, phosphorus, and potassium fertilizers, leads to the appearance of a lack of certain micro-elements in many countries of the world [45,46]. Banziger and Long [33] grew more than 1400 improved maize genotypes and 400 landraces in 13 trials in Mexico and Zimbabwe and found genotype differences in the contents of grain Fe and Zn. Maziya-Dixon et al. [47] observed a large variation in the contents of grain Fe and Zn in a set of 109 inbred lines that were developed for the mid-altitude and lowland agroecology of West and Central Africa. Carolina Feitosa de Vasconcelos et al. [48] studied the distribution of Zn in maize plants depending on the mode of application of Zn. Zinc was administered in the form of zinc sulfate (ZnSO_4_) over the soil (at doses of 0, 10, 20, 40, and 80 mg dm^−3^) and foliar at the third and fifth weeks after sowing (at concentrations of 0, 5, 10, 15, and 20 g L^−1^). The obtained results of the study showed that the Zn concentration in the roots and shoots of maize plants was increased by the application of Zn both over the soil and foliar over the leaves. In terms of the application form, the concentrations of Zn in the roots and shoots of plants showed a similar trend. The highest doses of Zn influenced the maximum concentration of Zn in plants in both modes of administration. A dose of 10 mg dm^−3^ of Zn applied over the soil caused a 72% increase in Zn concentration in shoots compared to the control variant, whereas by the foliar application of a 5 g L^−1^ dose, this increase was 67%. The concentration of Zn in the root increased by 88% by the application of 10 mg dm^−3^ of Zn across the soil, while doses of 20, 40, and 80 mg dm^−3^ led to an increase of 38, 34, and 28%, respectively. The Zn contents in maize plants ranged from 25 to 150 mg kg^−1^ in shoots, depending on the aeration and soil temperature, the humidity level in the root system zone, and the genotype of the cultivated plant. Although some authors point out the greater efficacy of foliar application over soil application, the adoption of Zn over the root has shown greater efficacy in these studies [49]. Starting from the assumption that the highlighting of genotypic differences regarding the concentration of elements in the grain may depend on external factors, as well as the cultivation technology used, primarily fertilization, Feil et al. [50], on the example of two tropical maize hybrids with the same genetic potential for fertility but with pronounced differences in the concentration of N, P, and K in the grain, showed that the concentration of individual elements was partly dependent on the carbohydrate content of the grain. In a three-year study with four genotypes of maize under drought conditions and irrigation throughout vegetation and the application of three different nitrogen fertilization levels, Feil et al. [51] found that the concentration of P, K, Ca, Mg, Zn, and Cu in the grain was quite stable, regardless of the treatments mentioned. The highest-yielding hybrid had the lowest concentration of most of the analyzed elements, so the above authors believed that there are still indications that high-yield breeding reduces the mineral content of the grain. Banziger and Long [33] stated that maize is a plant species of very high genetic fertility potential and that maize grain contains a higher concentration of trace elements than the grain of other high-yielding cereals but that the chances of increasing their content in the selection process within the existing elite germplasm are lower than with wheat and rice. Most studies to date of genetic variability in the content of individual elements that could be used in breeding programs have been limited to tropical maize germplasm, while very little data on similar studies are available for temperate maize [52]. Drought-tolerant genotypes with improved grain quality would represent a good starter base for various maize breeding programs. Given that Serbia is a large producer of corn, the identification and development of genotypes carrying desirable characteristics of grain quality would significantly expand the use value of corn, which would enable the greater placement of these products on the foreign market.

## 4. Materials and Methods

### 4.1. Overview of Experimental Site Description and Treatments 

The experimental location captured a major maize production region. The experiment was conducted at the Institute of Field and Vegetable Crops, National Institute of the Republic of Serbia situated at N 45° 19′, E 19° 50′. The experiment was set up on the chernozem soil, which belongs to automorphic soil types, class A–C (humus–accumulative soil, the subtype of chernozem on loess and loess-like sediments, the carbonate chernozem variety, medium depth) (IUSS Working Group WRB 2014; World Reference Base for Soil Resources 2014) and it is typical of the region where maize is intensely cultivated. This is a maize production area, with typical spells of drought during the growing season (Table 6).

Four divergent maize cultivars NS-4023, NS-6010, NS-6030, and NS-640 were grown under eight N combinations: fertilizer combinations with N addition in autumn and spring. The following factors were studied: N_1_: P_60_K_60_; N_2_: P_60_K_60_ + N_min spring_; N_3_: P_60_K_60_ + N_40autumn_ + N_min spring_; N_4_: P_60_K_60_ + N_60spring_; N_5_: P_60_K_60_ + N_100spring_; N_6_: P_60_K_60_ + N_40autumn_ + N_60spring_ + Zn; N_7_: P_60_K_60_ + N_40autumn_ + N_80spring_ + Zn; N_8_: P_60_K_60_ + N_160spring_ + Zn in both years of study. Zinc was applied as zinc sulfate (ZnSO_4_) in the amount of 1.0 kg ha^–1^ with foliar spraying, in the fourth and sixth week after sowing.

### 4.2. Soil Characteristics

Soil samples were collected from a 0–30 cm depth before sowing and some physiochemical characteristics were analyzed in the Research Laboratory, IFVCNS during both seasons. Organic matter was determined by the modified Walkley–Black method as suggested by Nelson and Sommers [53]. Available phosphorus (P) and potassium (K) were determined by the method of Olsen and Sommers [54]. Nitrogen was estimated according to Van Reeuwijk [55]. Before the experimentation, the soil samples were taken at a soil depth of 30 cm with an auger (end of March 2011), and the soil analysis report showed that the total soil N was (0.26 g kg^−1^), P_2_O_5_ (24.95 mg kg^−1^), and K_2_O (27.35 mg kg^−1^).

### 4.3. Agronomic Management

Winter wheat (*Triticum aestivum* L.) was the previous crop. Selected plots were plowed in October up to a 27–30 cm depth, and seedbed preparation was conducted before sowing with heavy-duty cultivators (Multi-Tiller) to a 15 cm depth in March. The crop was sown on 10 April 2011 and 18 April 2012 using a Wintersteiger AG pneumatic precision seed drill to a depth of 5 cm. The plot dimensions were 5 × 2.8 m, having an intra-row spacing of 22 cm and a row spacing of 70 cm. In both years, weed control was carried out by conventional chemical methods. Weed control consisted of a pre-emergence application of an S-metolachlor (960 g L^−1^) dose of 1.4 l ha^−1^ (2-chloro-N-(2-ethyl-6-methylphenyl)-N-(2-methoxy-1-methylethyl)acetamide) and 375 g L^−1^ S-metolachlor + 125 g L^−1^ Terbuthylazine + 37.5 g L^−1^ Mesotrione at a rate of 3.5 L ha^−1^ and a post-emergence application. During the vegetation season, *Sorghum halepense* sp. and other narrow-leaved weeds were controlled by applying Nicosulfuron or Rimsulfuron 50–60 g ha^−1^. During the vegetation season of maize, inter-row cultivation was carried out two times at the 3–5 leaf stage and 5–7 leaf stage.

### 4.4. Plant Sampling and Analysis

Plant tissue analyses included contents of N, P, K, Na, Ca, Mg, Zn, Mn, and Cu in the leaves and grain. Leaf samples (25 leaves) were taken under the cob in the silking stage (the second half of July). The samples were washed with deionized water, dried at 60 °C for 72 h, and ground using a silica grinder to pass a 0.5 mm sieve. After, maize harvests from each elementary plot cob were taken for grain chemical analysis. Samples of the plant material (leaves and grain) were prepared and milled in a mill for plant material grinding. Using AOAC Official method 972.43:2000 [56], the total N was determined. Using ICP-AES, the macro- and micro-elements were determined. Varian Vista-PRO Simultaneous ICP-AES, with axially mounted plasma, was used for these measurements. The iron content in maize grain was determined using atomic absorption spectrophotometry. The two center rows were used to collect yield data following Wasaya et al. [57], and the two adjacent rows were used for plant sampling. Grain yield per hectare was calculated on a 14% moisture basis and expressed in t ha^–1^.

### 4.5. Climatic Data

The climate of Serbia is moderate continental, with more or less pronounced local characteristics and a gradual transition between seasons. Weather characteristics, mainly precipitation and temperature regimes, are important factors of maize crop yields. 

Meteorological data including temperature and total precipitation were obtained from the automatic weather station of the Rimski Sancevi Agrometeorological Experimental Station (Figure 4). The area has a warm temperate continental climate, with an annual average temperature of 12.2 °C, and a mean annual precipitation of 528 mm. In both research years, the mean monthly air temperature increased from the beginning of June to the end of September. In 2011, the average monthly air temperature during the corn growing season was 1.4 °C higher than the multi-year average, while in 2012, this value was 2.5 °C higher. In 2011, the total amount of precipitation in the corn growing season was only 210.5 mm, which was 161.1 mm or 43.4% lower compared to the multi-year average. Similar values of the total amount of precipitation were noted in 2012. Compared to the multi-year average, the total amount of precipitation in this period was lower by 144.8 mm, i.e., by 39%. Extremely low amounts of precipitation in the growing season of 2011 and 2012 were followed by very high temperatures and dry and hot winds, which significantly affected the yield reduction. 

High temperatures associated with other stress factors affect the reduction in soil moisture, the occurrence of soil and air drought, the damage of reproductive organs, accelerated plant senescence, and yield reduction.

### 4.6. Statistical Analysis

An analysis of variance (ANOVA) test was applied to find the effects of treatments (T), cultivars (C), and T × C interactions on nutrient and elemental variables of leaf and grain for each of 2011 and 2012 years. Independ sample *t*-test was used to study the significance of variance of nutrient and elemental components in leaf and grain samples between the study years to compute the effects of changes in climatic conditions. Regression plots were developed to find the effects of leaf NPK contents on grain NPK contents. We applied Pearson’s correlation coefficients to analyze the relationships of leaf and grain NPK contents with leaf and grain elemental components (Na, Ca, Mg, Zn, Mn, and Cu). All data were analyzed using SPSS for Windows software v.19 [58].

## 5. Conclusions

We observed in our study that fertilization rates had different effects on the uptake and concentrations of the evaluated mineral nutrients. The environment in which the study was conducted, the management practices of the crop, soil conditions, and the growth stage of the crop all had significant effects on the observed differences in plant mineral nutrient contents. Evaluating changes in the accumulation of grain minerals across different genotypes can provide valuable information for the development of nutrient-enriched maize varieties. These results implied that maize yield was significantly affected by changes in cultivars and environments (mainly climatic changes from years). According to the results of the recent field experiments, nitrogen treatments (levels and splits) increase the production and quality of hybrid maize. Combining P_60_K_60_ + N_40autumn_ + N_60spring_ + Zn or P_60_K_60_ + N_40autumn_ + N_80spring_ + Zn tends to increase maize hybrid output. Final maize yield is a result of the interactions of genetic, environmental, and agronomic management factors. These findings suggested that Serbian farmers could improve maize hybrid yield performance by selecting the appropriate nitrogen fertilizer quantity and timing to build a more effective farming cycle with an environmentally friendly or more sustainable system. Future studies could evaluate more extreme plant densities, N fertilizer levels, and environments to further enhance our understanding of management effects on the mineral composition and maize yield in calcareous soil.

## Figures and Tables

**Figure 1 plants-13-00844-f001:**
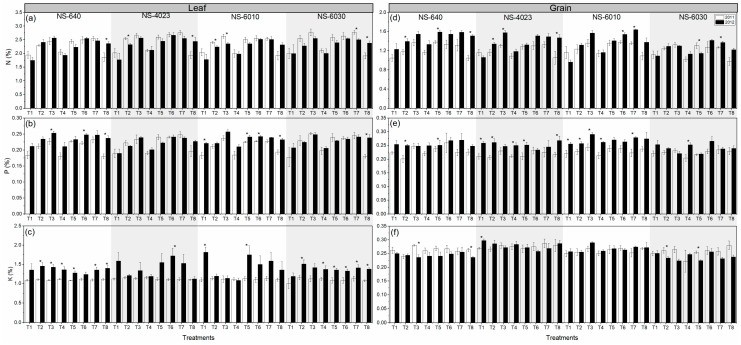
Fertilization effects on leaf (**a**) nitrogen (N), (**b**) phosphorus (P), and (**c**) potassium (K) contents and grain (**d**) nitrogen (N), (**e**) phosphorus (P), and (**f**) potassium (K) contents of four maize cultivars over two years. Bars are means of four replicates and contain standard errors of means (n = 4). Each pair of bars with asterisk (*) show significant differences between years at *p* < 0.05. T_1_: P_60_K_60_; T_2_: P_60_K_60_ + N_min spring_; T_3_: P_60_K_60_ + N_40autumn_ + N_min spring_; T_4_: P_60_K_60_ + N_60spring_; T_5_: P_60_K_60_ + N_100spring_; T_6_: P_60_K_60_ + N_40autumn_ + N_60spring_ + Zn; T_7_: P_60_K_60_ + N_40autumn_ + N_80spring_ + Zn; T_8_: P_60_K_60_ + N_160spring_ + Zn.

**Figure 2 plants-13-00844-f002:**
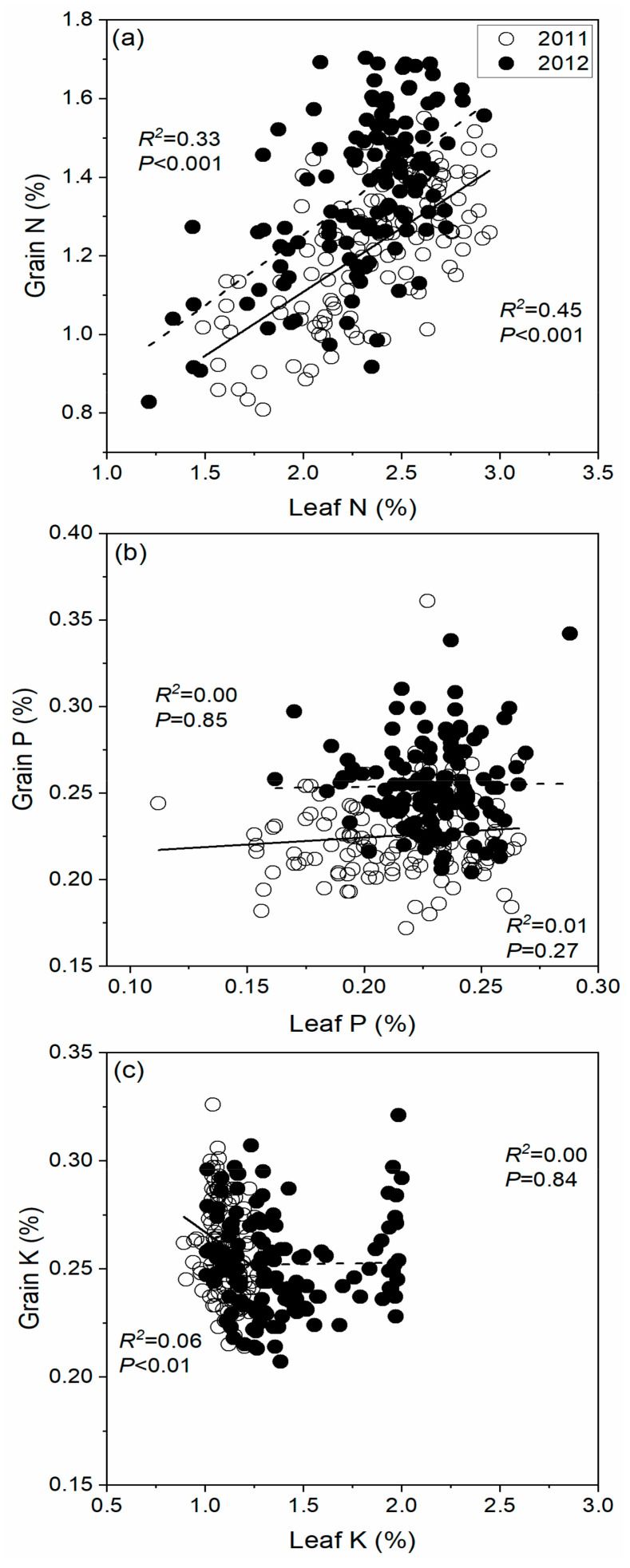
Relationships between leaf and grain (**a**) nitrogen (N), (**b**) phosphorus (P), and (**c**) potassium (K) contents for the two years.

**Figure 3 plants-13-00844-f003:**
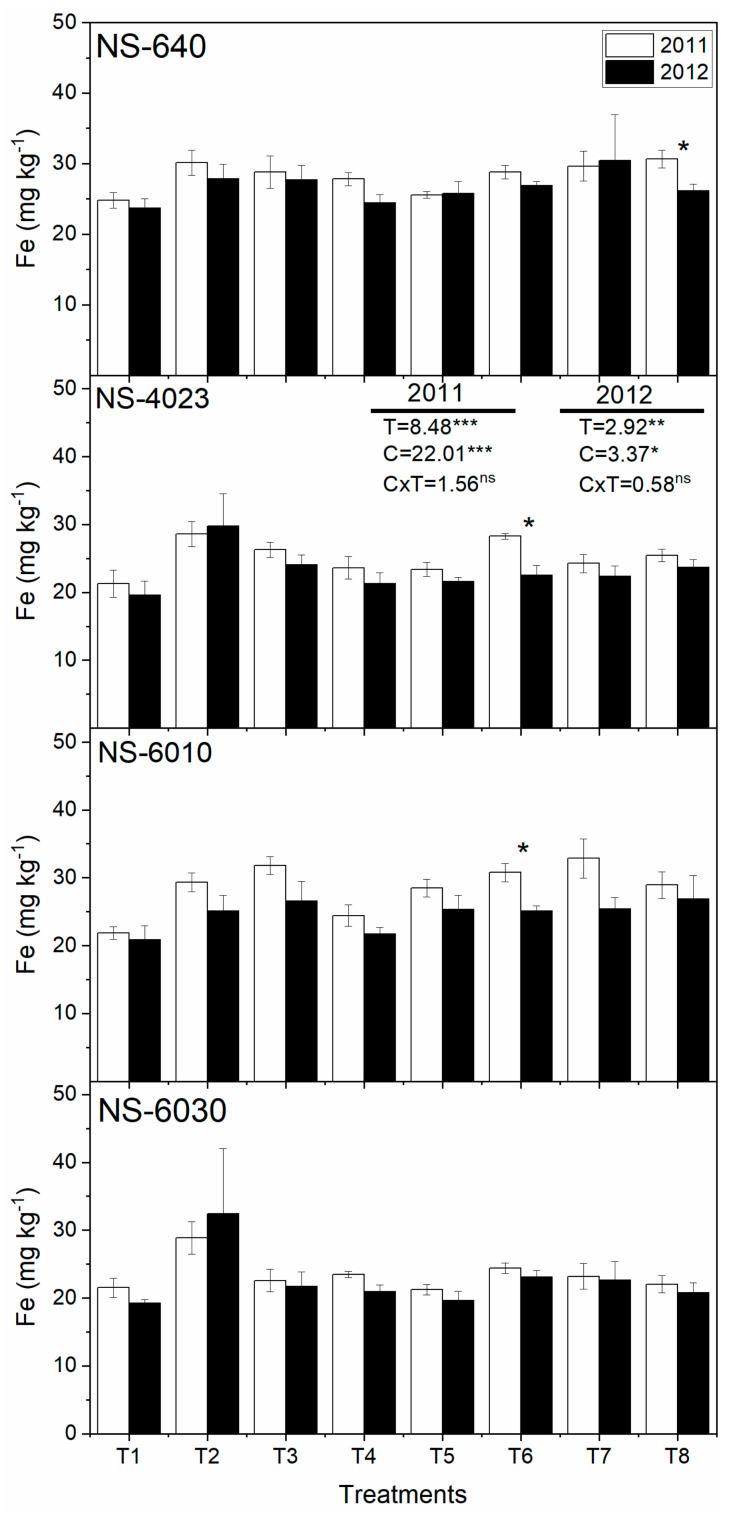
Fertilization effects on grain Fe contents of four maize cultivars over two years. Bars are means of four replicates and contain standard errors of means (n = 4). Each pair of bars with asterisk (*) show significant differences between years at *p* < 0.05. For ANOVA analyses, *, **, and *** show significant N fertilization (N), maize cultivar (C), and their interaction effects (N × C) at *p* < 0.05, *p* < 0.01, and *p* < 0.001, ns—not significant, respectively.

**Figure 4 plants-13-00844-f004:**
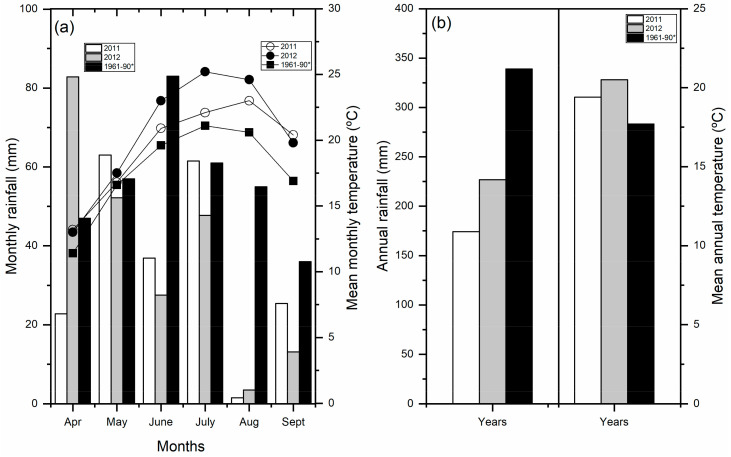
Climatic data on (**a**) monthly rainfall (mm) and mean monthly temperature (°C) and (**b**) annual rainfall (mm) and mean annual temperature (°C) for the experimental period during 2011 and 2012, and also historical data for years 1961-90. * long-term average.

**Table 1 plants-13-00844-t001:** *F*-statistics for the effects of N fertilization (N), cultivars (C), and their interactions (N × C) on leaf and grain N, P, and K contents.

Source	Leaf						Grain					
	Nitrogen (N)		Phosphorus (P)		Potassium (K)		Nitrogen (N)		Phosphorus (P)		Potassium (K)	
	1st Year	2nd Year	1st Year	2nd Year	1st Year	2nd Year	1st Year	2nd Year	1st Year	2nd Year	1st Year	2nd Year
N fertilization (N)	34.01 ***	21.48 ***	22.06 ***	20.10 ***	0.59 ^ns^	1.83 ^ns^	11.49 ***	32.76 ***	2.64 *	0.39 ^ns^	1.36 ^ns^	0.55 ^ns^
Cultivar (C)	3.94 *	1.39 ^ns^	2.74 *	5.60 **	0.56 ^ns^	0.47 ^ns^	1.37 ^ns^	22.33 ***	1.85 ^ns^	8.01 ***	0.50 ^ns^	16.11 ***
N × C	0.29 ^ns^	0.39 ^ns^	0.36 ^ns^	0.56 ^ns^	0.23 ^ns^	1.94 *	0.29 ^ns^	2.05 *	0.53 ^ns^	1.34 ^ns^	0.54 ^ns^	1.60 ^ns^

* *p* < 0.05, ** *p* < 0.01, and *** *p* < 0.001; ns = non-significant.

**Table 2 plants-13-00844-t002:** Fertilization effects on leaf micro-elemental contents of four maize genotypes over two years.

N Fertilization	Na (mg kg^−1^)								Ca (mg kg^−1^)								Mg (mg kg^−1^)							
	NS-640		NS-4023		NS-6010		NS-6030		NS-640		NS-4023		NS-6010		NS-6030		NS-640		NS-4023		NS-6010		NS-6030	
	1st Year	2nd Year	1st Year	2nd Year	1st Year	2nd Year	1st Year	2nd Year	1st Year	2nd Year	1st Year	2nd Year	1st Year	2nd Year	1st Year	2nd Year	1st Year	2nd Year	1st Year	2nd Year	1st Year	2nd Year	1st Year	2nd Year
N1	617	254	641	256	621	257	583	262	**0.93**	**0.76**	0.88	**0.69**	0.94	0.83	0.78	0.77	0.28	0.24	0.25	0.21	0.28	0.31	0.21	0.25
N2	598	274	669	266	594	263	665	273	0.89	0.85	0.82	0.77	0.85	0.87	0.84	0.83	0.28	0.29	0.24	0.24	**0.27**	**0.34**	0.23	0.28
N3	691	249	663	269	595	245	677	275	0.99	0.93	0.88	0.83	**0.94**	**0.88**	0.89	0.88	0.32	0.31	0.27	0.28	**0.31**	**0.38**	**0.26**	**0.30**
N4	596	270	656	265	601	280	629	262	0.87	0.84	0.83	0.74	0.89	0.85	0.81	0.83	0.28	0.27	0.24	0.23	**0.27**	**0.34**	0.23	0.28
N5	630	272	684	235	639	265	686	279	1.00	0.91	**0.97**	0.78	0.97	0.91	0.91	0.90	0.33	0.32	0.29	0.26	**0.31**	**0.38**	0.26	0.32
N6	682	254	703	269	665	261	675	293	1.01	0.93	0.93	0.83	0.96	0.91	0.89	0.92	0.33	0.32	0.28	0.30	**0.32**	**0.40**	**0.26**	**0.33**
N7	639	273	684	277	655	265	681	277	0.95	0.94	0.88	0.84	0.93	0.87	0.85	0.87	0.32	0.32	0.26	0.28	0.31	0.34	0.26	0.32
N8	672	274	691	249	698	248	710	275	**0.92**	**0.85**	0.86	0.74	**1.00**	**0.86**	0.85	0.85	0.27	0.30	0.24	0.23	0.28	0.36	0.23	0.31
ANOVA	1st Year: N = 1.49 ^ns^, C = 1.34 ^ns^, N × C = 0.26 ^ns^ 2nd Year: N = 1.61 ^ns^, C = 98.64 ***, N × C = 0.60 ^ns^								1st Year: N = 3.14 **, C = 6.93 ***, N × C = 0.36 ^ns^ 2nd Year: N = 11.45 ***, C = 24.19 ***, N × C = 0.70 ^ns^								1st Year: N = 2.67 *, C = 9.04 ***, N × C = 0.07 ^ns^ 2nd Year: N = 14.66 ***, C = 71.58 ***, N × C = 0.84 ^ns^							
	**Zn (mg kg^−1^)**								**Mn (mg kg^−1^)**								**Cu (mg kg^−1^)**							
	**NS-640**		**NS-4023**		**NS-6010**		**NS-6030**		**NS-640**		**NS-4023**		**NS-6010**		**NS-6030**		**NS-640**		**NS-4023**		**NS-6010**		**NS-6030**	
	**1st Year**	**2nd Year**	**1st Year**	**2nd Year**	**1st Year**	**2nd Year**	**1st Year**	**2nd Year**	**1st Year**	**2nd Year**	**1st Year**	**2nd Year**	**1st Year**	**2nd Year**	**1st Year**	**2nd Year**	**1st Year**	**2nd Year**	**1st Year**	**2nd Year**	**1st Year**	**2nd Year**	**1st Year**	**2nd Year**
N1	37.5	33.8	32.1	29.8	61.9	41.3	48.5	36.3	78.0	57.4	66.5	39.1	78.8	66.0	59.7	65.5	15.1	9.4	14.7	10.4	16.1	11.2	13.1	10.3
N2	44.1	40.2	36.0	35.2	41.5	39.2	39.8	36.7	77.5	72.9	66.5	50.9	78.4	75.1	68.0	78.1	15.7	12.6	17.7	11.8	15.6	12.3	17.9	12.3
N3	56.0	40.4	37.6	38.5	44.2	40.5	46.7	39.1	88.4	87.1	75.6	61.9	88.9	87.9	75.5	93.5	18.4	13.1	16.9	13.2	17.0	14.5	18.8	13.8
N4	30.4	31.2	38.3	34.9	48.3	35.0	36.7	36.0	72.2	64.7	64.6	44.6	74.4	72.4	61.9	73.0	15.0	11.1	15.1	11.2	16.7	12.4	15.0	10.8
N5	48.3	38.1	41.9	35.6	50.1	49.9	37.6	29.6	86.4	82.8	89.7	55.6	85.8	85.9	73.9	91.1	16.2	14.0	16.2	11.9	18.9	12.8	16.7	13.0
N6	58.5	59.6	35.9	38.1	59.0	48.3	35.5	46.6	88.9	90.5	77.6	58.5	87.9	86.9	75.1	93.0	19.7	13.0	18.4	13.9	19.9	13.1	18.4	13.9
N7	65.1	72.0	64.8	60.2	56.7	59.6	37.3	46.8	90.9	89.5	77.2	66.5	83.5	78.4	75.0	92.5	18.0	12.8	24.1	12.9	18.4	13.1	18.2	13.0
N8	54.5	43.7	33.9	37.2	36.5	35.1	26.8	30.4	74.9	74.9	60.7	51.3	73.6	78.1	58.1	87.1	20.4	13.3	15.0	12.0	15.7	13.0	15.2	12.7
ANOVA	1st Year: N = 1.69 ^ns^, C = 3.46 *, N × C = 0.95 ^ns^ 2nd Year: N = 29.50 ***, C = 10.14 ***, N × C = 3.14 ***								1st Year: N = 2.67 *, C = 4.13 **, N × C = 0.11 ^ns^ 2nd Year: N = 17.20 ***, C = 69.48 ***, N × C = 0.55 ^ns^								1st Year: N = 4.01 **, C = 0.27 ^ns^, N × C = 1.10 ^ns^ 2nd Year: N = 7.20 ***, C = 0.74 ^ns^, N × C = 0.47 ^ns^							

Values of each variable are means of four replicates. For each variable under each maize cultivar, bold values indicate significant differences between years at *p* < 0.05. For ANOVA analyses, *, **, and *** show significant N fertilization (N), maize cultivar (C), and their interaction effects at *p* < 0.05, *p* < 0.01, and *p* < 0.001, ns—not significant, respectively.

**Table 3 plants-13-00844-t003:** Fertilization effects on grain micro-elemental contents of four maize cultivars over two years.

N Fertilization	Na (mg kg^−1^)								Ca (mg kg^−1^)								Mg (mg kg^−1^)							
	NS-640		NS-4023		NS-6010		NS-6030		NS-640		NS-4023		NS-6010		NS-6030		NS-640		NS-4023		NS-6010		NS-6030	
	1st Year	2nd Year	1st Year	2nd Year	1st Year	2nd Year	1st Year	2nd Year	1st Year	2nd Year	1st Year	2nd Year	1st Year	2nd Year	1st Year	2nd Year	1st Year	2nd Year	1st Year	2nd Year	1st Year	2nd Year	1st Year	2nd Year
N1	155	168	153	169	156	168	149	172	0.13	0.15	0.13	0.15	0.13	0.15	0.13	0.16	0.11	0.13	0.12	0.13	0.12	0.12	0.12	0.13
N2	155	162	132	178	147	161	163	169	0.13	0.15	0.12	0.16	0.13	0.15	0.14	0.15	0.11	0.13	0.11	0.13	0.12	0.13	0.12	0.13
N3	170	152	157	155	168	166	166	168	0.14	0.14	0.14	0.14	0.14	0.15	0.14	0.15	0.13	0.12	0.12	0.13	0.13	0.14	0.13	0.12
N4	178	167	172	166	176	169	179	160	0.15	0.15	0.14	0.15	0.16	0.15	0.15	0.15	0.12	0.13	0.12	0.13	0.12	0.13	0.12	0.13
N5	263	168	158	166	158	166	172	173	0.20	0.15	0.13	0.15	0.13	0.15	0.14	0.16	0.14	0.13	0.12	0.13	0.12	0.14	0.12	0.12
N6	156	164	178	160	165	183	163	169	0.13	0.15	0.15	0.14	0.14	0.16	0.14	0.15	0.12	0.13	0.13	0.12	0.12	0.13	0.13	0.14
N7	177	170	166	164	155	163	144	163	0.15	0.15	0.13	0.15	0.13	0.15	0.12	0.15	0.12	0.13	0.12	0.13	0.12	0.14	0.13	0.13
N8	187	174	179	168	179	170	159	163	0.16	0.16	0.15	0.15	0.15	0.15	0.14	0.15	0.12	0.13	0.12	0.14	0.12	0.14	0.12	0.13
ANOVA	1st Year: N = 1.02 ^ns^, C = 0.99 ^ns^, N × C = 0.57 ^ns^ 2nd Year: N = 1.31 ^ns^, C = 0.45 ^ns^, N × C = 1.41 ^ns^								1st Year: N = 1.02 ^ns^, C = 0.93 ^ns^, N × C = 0.60 ^ns^ 2nd Year: N = 1.46 ^ns^, C = 1.22 ^ns^, N × C = 0.97 ^ns^								1st Year: N = 1.55 ^ns^, C = 0.34 ^ns^, N × C = 0.60 ^ns^ 2nd Year: N = 0.75 ^ns^, C = 1.60 ^ns^, N × C = 1.24 ^ns^							
	**Zn (mg kg^−1^)**								**Mn (mg kg^−1^)**								**Cu (mg kg^−1^)**							
	**NS-640**		**NS-4023**		**NS-6010**		**NS-6030**		**NS-640**		**NS-4023**		**NS-6010**		**NS-6030**		**NS-640**		**NS-4023**		**NS-6010**		**NS-6030**	
	**1st Year**	**2nd Year**	**1st Year**	**2nd Year**	**1st Year**	**2nd Year**	**1st Year**	**2nd Year**	**1st Year**	**2nd** **Year**	**1st** **Year**	**2nd Year**	**1st** **Year**	**2nd Year**	**1st** **Year**	**2nd Year**	**1st** **Year**	**2nd** **Year**	**1st** **Year**	**2nd** **Year**	**1st** **Year**	**2nd** **Year**	**1st Year**	**2nd Year**
N1	22.3	22.9	23.4	23.6	22.8	21.9	24.6	25.9	5.56	6.11	6.25	5.78	6.46	6.18	6.34	8.24	3.71	4.01	3.70	4.20	3.75	3.61	3.83	5.34
N2	37.7	23.2	32.3	23.6	21.7	22.2	29.6	23.6	5.57	6.41	6.37	6.17	6.80	7.62	6.26	8.51	4.90	4.06	4.54	5.10	3.64	4.56	5.63	5.22
N3	29.6	22.4	21.0	22.3	23.2	25.5	22.7	21.9	6.63	6.49	6.83	6.55	6.81	7.61	6.88	7.65	5.00	4.00	3.73	4.72	3.94	5.01	3.97	4.36
N4	23.7	22.9	28.8	24.5	32.5	21.6	22.8	24.5	5.93	6.12	7.05	5.71	6.32	7.03	5.94	8.24	3.80	4.89	4.26	4.18	5.48	4.28	3.89	4.10
N5	27.2	22.0	20.2	25.4	27.4	22.3	30.7	21.5	6.74	6.46	6.64	6.02	6.64	7.12	6.26	7.34	5.73	4.07	3.91	4.05	4.34	4.91	5.33	4.75
N6	30.6	23.5	25.5	21.6	20.7	22.8	27.3	24.9	5.86	6.69	6.88	6.08	6.14	7.41	7.30	8.33	4.02	4.38	5.65	3.84	3.72	5.80	4.31	4.25
N7	44.1	23.7	23.2	22.6	37.4	24.4	51.8	21.0	5.93	6.92	6.47	6.36	6.36	7.66	6.94	7.75	6.41	6.21	4.22	3.74	4.43	4.66	4.29	4.49
N8	21.2	23.0	21.9	25.3	33.4	25.1	21.7	22.5	5.99	6.51	6.63	6.70	6.70	7.57	6.17	7.83	4.18	5.70	4.14	4.33	7.27	7.02	3.59	4.67
ANOVA	1st Year: N = 2.06 ^ns^, C = 0.77 ^ns^, N × C = 0.72 ^ns^ 2nd Year: N = 0.20 ^ns^, C = 0.19 ^ns^, N × C = 0.86 ^ns^								1st Year: N = 0.86 ^ns^, C = 3.10 *, N × C = 0.66 ^ns^ 2nd Year: N = 1.54 ^ns^, C = 35.61 ***, N × C = 0.85 ^ns^								1st Year: N = 0.53 ^ns^, C = 0.30 ^ns^, N × C = 0.84 ^ns^ 2nd Year: N = 1.19 ^ns^, C = 1.83 ^ns^, N × C = 1.54 ^ns^							

Values of each variable are means of four replicates. For each variable under each maize cultivar. For ANOVA analyses, * and *** show significant N fertilization (N), maize cultivar (C), and their interaction effects at *p* < 0.05, and *p* < 0.001, ns—not significant, respectively.

**Table 4 plants-13-00844-t004:** Pearson’s correlation coefficients for the relationships between leaf and grain N, P, and K contents with leaf and grain micro-elemental contents over two years.

Year	Variable	Leaf Na	Leaf Ca	Leaf Mg	Leaf Zn	Leaf Mn	Leaf Cu	Grain Na	Grain Ca	Grain Mg	Grain Zn	Grain Mn	Grain Cu	Grain Fe
1st Year	Leaf N	0.23 **	0.22 *	0.30 **	0.09	0.26 **	0.44 **	0.01	−0.04	0.23 *	0.08	0.14	−0.05	0.15
	Leaf P	0.22 *	0.43 **	0.39 **	0.15	0.42 **	0.43 **	0.04	−0.01	0.16	0.14	0.11	0.10	0.05
	Leaf K	0.44 **	0.22 *	0.15	−0.11	0.40 **	0.06	−0.11	−0.10	−0.11	0.09	0.04	0.15	0.10
	Grain N	0.29 **	0.36 **	0.41 **	0.20 *	0.26 **	0.37 **	0.02	−0.02	0.28 **	0.13	0.19 *	0.02	0.21 *
	Grain P	0.05	0.16	0.13	0.09	0.11	0.04	0.10	0.11	0.60 **	−0.05	0.05	−0.02	0.18 *
	Grain K	0.14	0.08	0.13	0.01	−0.07	0.11	0.18 *	0.18 *	0.56 **	−0.01	0.12	0.01	0.11
2nd Year	Leaf N	0.20 *	0.37 **	0.35 **	0.32 **	0.44 **	0.60 **	−0.10	−0.18 *	0.11	−0.03	0.09	0.07	0.11
	Leaf P	0.08	0.49 **	0.51 **	0.40 **	0.63 **	0.59 **	−0.09	−0.10	0.06	−0.08	0.14	0.07	0.22 *
	Leaf K	0.04	−0.04	0.08	0.06	0.15	−0.01	−0.06	0.04	0.05	−0.03	0.04	−0.10	0.04
	Grain N	−0.02	0.40 **	0.36 **	0.46 **	0.34 **	0.45 **	−0.02	−0.13	0.40 **	0.06	−0.03	0.12	0.32 **
	Grain P	−0.07	0.01	0.14	0.23 **	−0.03	−0.02	0.21 *	0.02	0.89 **	0.56 **	−0.02	0.11	0.39 **
	Grain K	−0.10	−0.12	0.08	0.08	−0.20 *	−0.08	0.27 **	0.10	0.81 **	0.58 **	−0.11	0.11	0.30 **

* *p* < 0.05 and ** *p* < 0.01; *n* = 128.

**Table 5 plants-13-00844-t005:** Fertilization effects on grain yield (kg ha^−1^) of four maize cultivars over two years.

N Fertilization	NS-640		NS-4023		NS-6010		NS-6030	
	1st Year	2nd Year	1st Year	2nd Year	1st Year	2nd Year	1st Year	2nd Year
N1	6376	5528	5625	4755	**6511**	**5670**	6503	6427
N2	**10,518**	**7168**	**9794**	**6837**	**10,024**	**7163**	**10,096**	**8006**
N3	**10,741**	**8357**	**10,993**	**7620**	**11,320**	**8028**	**11,785**	**8317**
N4	**9368**	**7304**	**10,038**	**6868**	**10,367**	**7121**	**10,226**	**8075**
N5	**10,733**	**7882**	**10,689**	**6981**	**11,092**	**7184**	**11,206**	**7716**
N6	**12,310**	**7991**	**11,231**	**7102**	**12,502**	**7773**	**12,163**	**7840**
N7	**12,934**	**7297**	**10,981**	**6214**	**12,044**	**7828**	**12,738**	**7084**
N8	**10,282**	**6707**	**9940**	**5887**	**10,826**	**6865**	**10,251**	**7020**
ANOVA	2011: N = 128.47 ***, C = 7.93 ***, N × C = 1.74 ^ns^; 2012: N = 29.57 ***, C = 19.68 ***, N × C = 1.17 ^ns^							

Values are means of four replicates. For each maize cultivar, bold values indicate significant differences between years at *p* < 0.05. For ANOVA analyses, *** show significant N fertilization (N), maize cultivar (C), and their interaction effects (N × C) at *p* < 0.001. ns = non-significant effects.

**Table 6 plants-13-00844-t006:** Overview of experimental site.

Site	Abbreviation	Soil Type	Long (°E)	Lat (°N)	Alt (m)
Novi Sad	NS	Calcareous chernozem	19.51	45.20	84

## Data Availability

Data will be made available on demand.

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
