# Peer review of "Nitrogen Fertilization and Cultivar Interactions Determine Maize Yield and Grain Mineral Composition in Calcareous Soil under Semiarid Conditions"

_plants, 2024, doi:10.3390/plants13060844_

Round 1
Reviewer 1 Report
Comments and Suggestions for Authors
The paper particularly focuses on the effect of N fertilization on maize cultivars in Serbia.
The “1. Introduction” is very short and limited. For example, previous studies on maize cultivars are not mentioned. On the other hand, the “3. Discussion” is mainly a literature review rather than discussing the results that were found.
Objectives: “…an overall objective to find the how N fertilization and cultivar interactions along climatic conditions…
Interannual climate variability is expected. However, the study only involves two years.
“2. Results”
Meaning of “T1”, “T2”….?
The data presented in the tables should not be repeated throughout the text (treatments could be compared in relative terms).
“4. Materials and Methods”
Table 6: Meanings of “AWC” and “Moderate”?
Information about the experimental design is missing.
Number of samples collected per plot?
“5. Conclusions”
What was the recommended N fertilization overall? And recommended cultivar?
Author Response
We are thankful to the Reviewer for his/her evaluation. We thank the reviewer for the favorable general comments. All given comments are addressed properly.

Reviewer 2 Report
Comments and Suggestions for Authors
the manuscript examines the interactions between genotypes of maize (four) and nitrogen (P and Zn) seasonal fertilisations. The experiments were done during two years. The topic is worthy for investigation and meets sstandard expectations of Plants.
nevertheless, the manuscript presents serious concerns.
1-lacks of literature background. For example, in introduction, nine references were used. This is nt sufficient for a species, which is, agriculturally, among the most studies crop species for N use and N use efficiency. Several reports have been performed to examine N use efficiency using isotopes (very performent tool). The introduction should be modified in order to state clearly the originality and noverlty of this work.
2-the results are presented as a succession of interactions and correlations instead of mechanistic approach. For example in fig. 2 plot 3 (G Kby Leaf K) for 2012 it seems that are two groups of points: right one ( seems 20 points) and the others. This fact was not discussed. If authors identified the points by their correspondance, It can be identify some tendancies for may be genotype of answer to climatic conditions or sensitivity to soil composition, or better for this manuscript N use and N use efficiency...This part needs deep modifications.
Secondary remarks
authors used four divergent genotypes. Divergent for what? do you mean contrasted for N use? for yield?
This point is important and should be highlighted along the manuscript.
Moreover, information concerning the four genetypes is lacking.
Title is too long
Abstrat
Please delete « in South Pannonian Basin ». Please change in arid conditions by under arid conditions
Please rewritte the conclusion to meet the objectives of the study
« This experiment was conducted with an overall objective to find the how N fertilization and cultivar interactions along climatic conditions determine mineral composition and maize yield responses of four divergent maize cultivars were grown under eight different fertili-zation levels. »
Please rephrase
Comments on the Quality of English Language
The manuscript needs English editing
Author Response

(The authors gave the same response as above.)

Reviewer 3 Report
Comments and Suggestions for Authors
Dear authors and editors,
the manuscript “Nitrogen fertilization and cultivar interactions determine maize yield and grain mineral composition in calcareous soil in semi-arid conditions in South Pannonian Basin” aims to investigate the interactions between N fertilisation and cultivar to determine maize yield and quality. I don’t think your results bring any news for the international community, as several studies have already been published on the same topic. Moreover, you present two-year research; a longer observation period would have provided more reliable results. Although the object of the study is not new, I appreciate your effort to provide some data for Serbian agriculture. However, the paper must be revised before it is considered for publication.
The Introduction is too short and not focused. You should present a state-of-the-art of what has been done in this field. There is plenty of literature about N fertilisation and cultivar effects on yield. Moreover, the objectives of the research must be clearly stated.
Although the Material and methods and the results are clearly explained, I can’t find information on Fe determination in the M&M, while data on Fe content are reported in the results.
I enjoyed reading the Discussion and your conclusions on the importance of your study.
Here are some specific comments:
- In the abstract, the sentence “This experiment was conducted with an overall objective to find how N fertilization and cultivar interactions along climatic conditions determine mineral composition and maize yield responses of four divergent maize cultivars were grown under eight different fertilization levels.” has some grammar issues that makes it hard to understand.
- There are several grammar or syntax mistakes throughout the text. I suggest going through a deep revision of the English editing.
Comments on the Quality of English LanguageSpelling, syntax and grammar mistakes have been detected
Author Response

(The authors gave the same response as above.)

Round 2
Reviewer 1 Report
Comments and Suggestions for Authors
The authors revised the manuscript satisfactorily.
Reviewer 3 Report
Comments and Suggestions for Authors
The authors addressed all my major comments. The manuscript quality has been improved